# Unveiling Niaprazine’s Potential: Behavioral Insights into a Re-Emerging Anxiolytic Agent

**DOI:** 10.3390/biomedicines12092087

**Published:** 2024-09-12

**Authors:** Hanna Trebesova, Martina Monaco, Sara Baldassari, Giorgia Ailuno, Edilio Lancellotti, Gabriele Caviglioli, Anna Maria Pittaluga, Massimo Grilli

**Affiliations:** 1Pharmacology and Toxicology Unit, Department of Pharmacy, University of Genova, 16148 Genoa, Italy; hanna.trebesova@unige.it (H.T.); martina.monaco@edu.unige.it (M.M.); annamaria.pittaluga@unige.it (A.M.P.); 2Pharmaceutical Technology Unit, Department of Pharmacy, University of Genova, 16148 Genoa, Italy; sara.baldassari@unige.it (S.B.); giorgia.ailuno@unige.it (G.A.); gabriele.caviglioli@unige.it (G.C.); 3Farmacia Assarotti, 16122 Genoa, Italy; info@farmaciaassarotti.it; 4IRCCS Ospedale Policlinico San Martino, 16132 Genova, Italy

**Keywords:** Niaprazine, anxiety, elevated plus maze, marble test, video tracking analysis, PCA

## Abstract

Ongoing global research actions seek to comprehensively understand the adverse impact of stress and anxiety on the physical and mental health of both human beings and animals. Niaprazine (NIA) is a chemical compound that belongs to the class of piperazine derivatives. This compound has recently gained renewed attention due to its potential therapeutic properties for treating certain conditions such as anxiety. Despite its potential benefits, the behavioral effects of NIA have not been thoroughly investigated. This study aimed to examine NIA’s potential as an anti-anxiety and anti-stress agent. After administering either vehicle or NIA in their drinking water to mice for 14 days, we conducted behavioral analyses using the Marble Burying Test and the Elevated Plus Maze test. NIA-treated mice spend more time in the open arms and bury fewer marbles. Moreover, a stability study confirmed the linear relationship between NIA concentration and its response across concentrations encompassing the NIA mother solution and the NIA solutions administered to mice. Also, a preliminary synaptic toxicity analysis showed no direct damage to cortical nerve endings. Here, we show that NIA can modulate anxiety-related behaviors without significantly impacting exploratory activity or adverse effects. Our work describes new findings that contribute to the research on safer and more tolerable anxiety management options.

## 1. Introduction

The global data on the impact of stress and anxiety on human and animal health are relentless [1]. Recent geopolitical, economic, social, and health problems have increased awareness that new impulses will necessarily need to be put in place to find increasingly effective remedies. Social interactions between individuals have significantly changed due to the proliferation of new technologies [2,3,4]. Conversely, modern society is increasingly characterized by direct interactions between humans and pets [5,6,7]. Therefore, evaluating the relationship between pets and people in civil society highlights how they can mutually influence each other during stressful events [8]. It is increasingly evident that the health of one subject can be negatively affected by the condition of the other, leading to a feedback loop [9,10]. Possible therapeutic strategies vary from lifestyle changes to psychological therapy and pharmacological treatments [11,12]. Anti-stress agents refer to a diverse range of pharmaceutical and natural substances utilized to alleviate stress and anxiety [13,14]. This category encompasses antidepressants prescribed to manage mood disorders [15,16,17], benzodiazepines, which function as tranquilizers with sedative effects [18,19,20], adaptogenic drugs recognized for their ability to help the body adapt to stress [21,22], and natural extracts known for their calming and stress-relieving properties [23,24].

In this field, the possible approaches are the identification of new active molecules that may be more effective and have fewer side effects or the re-evaluation of old drugs with undiscovered potential [25,26]. One such drug is Niaprazine (NIA), a piperazine derivative with antihistamine and sedative effects. NIA has been used in the past for sleep disorders in children [27]. Although it is not available in all countries and was withdrawn from the market in the United States due to the risk of liver damage, it is still available in other countries like France and Canada. In addition, antihistamines can cause serious anticholinergic side effects such as mydriasis, urinary retention, constipation, tachycardia, hypotension, dystonia, and sedation [28]. In terms of its pharmacodynamic profile, this drug has a low affinity for the vesicular monoamine transporter and the D2, α2, and H1 receptors. Conversely, it exhibits a higher affinity for α1 and 5-HT2 [29] but shows no affinity for 5-HT1A and 5-HT1B. NIA also induces a short-term depletion of noradrenaline (NA) and dopamine (DA) in the rat brain [30]. Its primary metabolite is p-fluoro-phenylpiperazine (FPP). FPP reduced 5-HIAA and 3,4-dihydroxyphenyl acetic acid (DOPAC) in the rat brain and inhibited the uptake of 5-HT and NA in vitro [30]. The reduction in catecholamine levels was transient and was accompanied by increased brain levels of the metabolites DOPAC and MOPEG-SO4. Unlike NIA, FPP did not induce behavioral sedation, but at high doses, it triggered a behavioral syndrome suggesting serotonergic stimulation [30,31,32]. While NIA has been reassessed for therapeutic properties in certain disorders [33,34,35], we aimed to explore its potential as an anti-stress agent. To investigate this, we conducted a behavioral analysis using mice treated with either a vehicle or NIA in their drinking water for 14 days. The results of our study showed that NIA can modulate aspects related to anxiety without significantly affecting exploratory activity in the animals.

## 2. Materials and Methods

### 2.1. Animals

Male C57BL/6J mice were purchased from Charles River (Calco, Italy). The mice were raised for up to 3 and 5 months in the animal facility of the Department of Pharmacy, Section of Pharmacology and Toxicology, School of Medical and Pharmaceutical Sciences, University of Genoa. The experimental procedures followed European legislation (Directive 2010/63/EU for animal experiments) and the ARRIVE guidelines [36]. They were approved by the Committee on the Ethics of Animal Experiments of the University of Genoa and the Italian Ministry of Health (DDL 26/2014 and previous legislation; permit numbers 50/2011-B and 612/2015-PR, authorization no. 75F11.N.JP6).

### 2.2. Behavioral Tests

In the present behavioral setup, mice were randomly divided into two groups before starting treatment: control and treated (NIA), a total of n = 14 mice (n = 6 controls; n = 8 treated mice). ARRIVE guidelines 2.0 were taken into account during all the steps of the experimentation phase [36]. Data collection was based on a pipeline where each mouse is characterized by a combination of standardized and validated tests (Elevated Plus Maze and Marble Test). A small but representative sample size was selected to meet the 3R principles in reduction on animals used for the experiment. To obviate a limited number of animals, many descriptive variables were collected thanks to the behavioral video analysis software. A total of 26 variables (motor parameters, frozen events, and anxiety-related variables) were extrapolated. The criteria for inclusion/exclusion were selected a priori. A possible exclusion criterion for EPM is the fall of the mouse during the test, while during the MT, the mouse should dig the bedding, which is a starting point of the burring event. All mice met the inclusion criteria in both EPM and MT. Due to behavioral tests and continuous treatment supply, the investigators could not be blind, but EPM and MT were performed with ToxTrac v024.1.2 video tracking software using an overhead video camera system to automate behavioral testing and provide unbiased data analyses. Mice activity was recorded as average speed, mobility and exploration rate, and total distance traveled. Time spent in the open arm during EPM was calculated from video analysis, and the number of marbles was counted at the end of each test.

#### 2.2.1. Elevated Plus Maze

The Elevated Plus Maze consists of a cross arena with 4 arms connected in the middle zone; each arm is 36 cm long and 6 cm wide, and the whole maze is 50 cm elevated from ground level [37,38]. The closed part of the cross has transparent plexiglass 25 cm high sidewalls that provide shelter to the animal. The video analysis phase recorded numerous parameters performed by both the operator and software. The ToxTrac v024.1.2 computer software was utilized to process and analyze the behavioral parameters gathered from the video recordings. The experiments were conducted using identical protocols for handling procedures, brightness levels, and timing (during the dark cycle) across all groups. Following each trial, the entire arena was thoroughly cleaned using 70% ethanol and allowed to evaporate [39].

#### 2.2.2. Marble Burying Test

The Marble Burying Test was conducted utilizing conventional polycarbonate cages with dimensions of 41 × 25 × 20 cm, wherein the floor was lined with 5 cm of sawdust. Subsequently, twenty marbles were evenly distributed within the arena, arranged in five rows and four columns [40]. A solitary mouse was placed in the center of the cage, where it had the freedom to explore for 10 min. After the testing time, each animal was returned to their respective home cage. The experiment only counted marbles that were buried 2/3 or more, while marbles buried less than 2/3 were not included. All trials were recorded on video and analyzed with ToxTrac v024.1.2 software [41]. Identical handling procedures, brightness levels, and timing (during the dark cycle) were followed across all groups during the experiments.

### 2.3. Flow Cytometry

To fulfill the 3Rs, a limited number of animals were sacrificed for the preparation of synaptosomes. Cortices were selected for ex vivo analysis to provide sufficient material to perform apoptotic assays.

Crude synaptosomes from the mouse cortex were prepared with slight modifications, as described previously [42,43,44]. Briefly, 100 µL aliquots were separated to label the synaptosomes. Viability after 1 h incubation with NIA was assessed using a Guava^®^ easyCyte™ flow cytometer (Luminex Corporation, Austin, TX, USA). Double staining with Annexin-V and Calcein-AM allowed the distinction of early apoptosis and viable neuronal particles, respectively. Annexin-V PE has a high specificity and affinity for phosphatidylserine (PS), which is translocated to the outer leaflet of the plasma membrane during early apoptosis [44,45,46,47,48]. To label the synaptosomes with Annexin-V, they were resuspended in 95 µL of Annexin-V Binding Buffer and then enriched with 5 µL of Annexin-V PE conjugate. The sample was incubated for 15 min in the dark at room temperature and then washed again with binding buffer. Meanwhile, a Calcein-AM working solution was prepared at a final concentration of 2 µM (by diluting a 2 mM Calcein-AM stock solution in PBS). An amount of 40 µL of the working solution was added to the sample and incubated at 37 °C for 30 min, followed by washing with MP [49]. Some samples were labeled with both Annexin-V and Calcein-AM to describe the ratio between apoptotic and non-apoptotic particles [45].

### 2.4. Chemical Analysis

#### 2.4.1. Stability Study

The stability of NIA aqueous solution administered to mice as drinking fluid was assessed by analytical reverse phase HPLC/UV, adapting the method reported by Fisichella et al. [50]. The analyses were carried out on a Hewlett Packard Series II 1090 liquid chromatograph (HP Italy S.r.l., Cernusco sul Naviglio, Italy) with a UV–visible detector equipped with HPLC ChemStation software Empower 3, using an RP18, X-Bridge C18 column, 3.5 µm, 4.6 × 150 mm, and a Waters X-Bridge BEH C18 Sentry Guard Cartridge, 3.5 µm, 4.6 × 20 mm (Waters, Milford, MA, USA). The isocratic mobile phase was composed of mQ water with 0.1% formic acid at pH 2.71 ± 0.02 (solvent A) and MeOH gradient grade with 0.1% formic acid (solvent B) in a 55:45 v/v ratio. The chromatograms were acquired at 228, 245, 200, and 302 nm wavelengths, with 228 nm as the analytical wavelength; the injection volume was 10 µL, and the flux was 0.8 mL/min.

Each analytical sample was analyzed as such, with no further dilution, in triplicate. To validate NIA response linearity, different concentrations of NIA ranging from 4.5 μM to 911 µM in mQ water were used. A standard curve was obtained by plotting NIA peak area against NIA concentration (Appendix A). Ordinary least squares regression was performed.

#### 2.4.2. Stress Test

To force NIA hydrolysis with the aim of identifying possible degradation products, a 2428 μM NIA solution in 0.1 N HCl aqueous solution (pH 1.23) and a 780 μM NIA solution in 0.1 N NaOH aqueous solution (obtained with the aid of 30 min sonication in an ultrasonic bath; pH 12.80) were prepared, transferred in a sealed vial, and heated at 60 °C in a glycerine bath for 5 h under stirring. The lower concentration of the alkaline solution was due to the reduced NIA solubility in the basic medium.

At the end, the solutions were cooled down, their pH was corrected to pH 5 (reaching final NIA concentrations of 2355 and 722 μM, respectively) and were analyzed with the HPLC method described above. In parallel, equivalent volumes of acidic and alkaline medium underwent the same treatment and were analyzed as blanks. As nicotinic acid is a possible NIA-related substance, a 2053 μM nicotinic acid solution in mQ water was also prepared and analyzed.

### 2.5. Chemicals

Niaprazine was kindly supplied by Farmacia Assarotti (16122, Genoa, Italy). Annexin-V was purchased from Miltenyi Biotec GmbH, Bergisch Gladbach, Germany. PE conjugate was used for all assays (order # 130-118-363). Calcein-AM was purchased from Biotium, Fremont, CA, USA (catalog no. 30026).

### 2.6. Data Analysis and Statistics

Univariate statistical analysis and corresponding graphical representation were conducted using Past 4.13 and R 4.4.0 [51]. Analysis of variance was performed by ANOVA followed by Tukey’s multiple comparison test. Data are presented as the mean ± standard error of the mean (SEM) and considered significant for *p* < 0.05 at least. Multivariate data analysis has been performed by Past 4.13 and JMP Pro 17.2.0. Principal Component Analysis (PCA) is a statistical procedure designed to reduce the dimensionality of data while preserving variance as much as possible. This is done by finding a new set of orthogonal variables called principal components (PCs), which are linear combinations of the original variables.

## 3. Results

Young male mice received a dose of 1 mg/kg/day of NIA in their drinking water. The solutions had a concentration of approx. 14.5 µM and were freshly prepared by diluting 1 mL of an 842 µM mother solution to 50 mL with drinking water; the drinking solutions were replaced every 4 days to reduce the stress caused to the animals and to minimize inter-day dosage variability. The administration lasted 14 days, during which the mother solution was stored at 2–8 °C. This dose was chosen based on previous studies and data in children [27,52]. This dosage has been reported to reduce mild anxiety signs both in children and adolescents without significant side effects [53]. We acknowledge that this administration method does not provide precise dosage information, as we would have obtained through gavage. However, we chose this method to minimize stress and avoid affecting the treatment responses of the animals [54]. To shed light on possible neurotoxic effects, we performed a cytofluorimetric analysis of apoptosis signals in cortical synaptosomes exposed to increasing concentrations of NIA. Previous studies have shown that marking synaptosomes with both Calcein-AM and Annexin-V can reveal signs of synaptic degeneration [42,48,55]. Figure 1A–C show that after 1 h of in vitro exposure to NIA concentrations ranging from 10 nM to 1 µM, the percentage of colocalization of Calcein-AM and Annexin-V remained stable, indicating no change in the apoptotic rate.

As shown in Figure 2A, body weight remained stable in both the control and treated animals, suggesting that NIA is well tolerated. The amount of water drunk was similar in both groups (Figure 2B). Daily animal observations revealed no obvious signs of toxicity, including apathy, weight changes, or secretions. We established recurring time points (Figure 3) to create a graph showing the weight progression.

The stability of the NIA mother solution and diluted solutions administered as drinking fluid to mice was assessed by HPLC. First, the linearity of the analyte response at 228 nm was confirmed in the range of 4.5–911 µM (Appendix A).

During the study, the drinking solutions were freshly prepared every 4 days for the 2-week duration of the experiment; therefore, samples of 14.5 μM NIA aqueous solutions were analyzed in triplicate after 4-day storage at room temperature. The chromatograms at all the registered wavelengths did not show any additional peak besides the one for NIA at approx. 5.5 min retention time (Figure 4); moreover, at all wavelengths, the injection peak, extending from 1.5 to 2.5 min, was not different from the one for the blanks, and the NIA concentration, calculated against the standard calibration curve, was unchanged.

To validate the manufacturing procedure involving the preparation of the 14.5 μM drinking solutions from an 842 μM NIA mother solution, the stability of the mother solution was also verified by analyzing it after 4 months of storage at 2–8 °C. Also, in this case, the chromatograms at all wavelengths showed no sign of alteration.

As Niaprazine is not reported in a European Pharmacopoeia monograph, no information about its degradation products is available, nor has any stability-indicating analytical method been published. To identify possible degradation products, we performed a stress test by exposing NIA to acidic and alkaline conditions. For this purpose, 2428 μM and 780 μM NIA solutions in 0.1 N HCl and 0.1 N NaOH aqueous solutions, respectively, were prepared. The concentration of the alkaline solution was close to the solubility limit of NIA at basic pH. After heating them at 60 °C for 5 h, the pH of both solutions was corrected to 5, and the solutions were analyzed in comparison to equal volumes of acidic and alkaline media that had undergone the same heat treatment and pH correction. No additional peaks were found in both chromatograms in comparison to the freshly prepared solutions, and NIA concentrations were undiminished if compared with the initial ones. Moreover, any hydrolysis was ruled out against the chromatogram of 2053 μM nicotinic acid, whose retention time (2.3 min) does not correspond to any peak in the chromatograms of the NIA solutions that underwent acidic and alkaline treatments.

At the end of the treatment, animal behavior was analyzed in two different test setups: Marble Test (MT) and Elevated Plus Maze (EPM). Each animal was individually recorded for ten minutes in both arenas without any interference from the operator. Using Principal Component Analysis (PCA), we analyzed seventeen variables, including motor and anxiety parameters.

The multivariate analysis displayed two distinct clusters representing the two types of oral treatments, as shown in Figure 5. PCA on the variable scores extracted three factors with an Eigenvalue greater than 2. Table 1 shows the component values. Specifically, components 1, 2, and 3 explain 65.2% of the variance.

Component 1 accounts for 37.6% of the variance. The variables that loaded positively on this factor were Average Speed, Mobility Average Speed, Average Acceleration, Mobility Rate, and Distance (Figure 6A,B). Component 2 accounts for 15.6% of the variance. The variables that ranked positively on this factor were Frozen Events, Total Time Frozen, Max Speed, Max Accel, and time center (Figure 6A,B). Component 3 accounted for 12% of the variance. The variables that impacted positively on this factor were Max Speed, Max Accel, marbles, time in Open Arm (OA), and time corner (Figure 6A,B).

Although the difference is not statistically significant, NIA produced a decrease in the distance covered. The mean distance covered was 24,554 ± 1754 mm with NIA, compared to 29,417 ± 3134 mm without NIA. The Average Speed (45.25 ± 3 vs. 52.92 ± 5), the Max Speed (907.08 ± 81 vs. 1027.76 ± 176), and the Average Acceleration (497.79 ± 21 vs. 562.67 ± 40) are slightly reduced in the NIA group.

These effects are supported by Figure 7, which shows the time-by-time speed profile of the two groups. Indeed, NIA increased the frequency in the low-speed range, further supporting the concept that it may reduce the motor activity of mice. We obtained further support for this hypothesis by analyzing the frequency and location of breaks taken by the animals during the experimental session.

NIA increased the number of frozen events and immobility time (23 ± 4 s vs. 16 ± 6 s (Figure 8A,B)). The software detected periods during which the animal traveled less than 3 mm for 5 s. Figure 9 shows a representative analysis of the localization of freezing events for control (left) and NIA-treated mice (right), respectively. The heatmaps reveal that while NIA induces a small increase in stops, the spatial distribution of these stops remains unchanged. Indeed, the peripheral zone continues to be preferred by the mice.

The sedative properties of NIA are widely known, and our data confirmed that the chosen dose affects mice. Interestingly, this oral antihistamine treatment maintains the exploration rate (percentage of sectors explored in the arena 86 ± 2% vs. 84 ± 1%), even with a slight, but not significant, decrease in the number of buried marbles (15.5 ± 1 vs. 18.7 ± 1) (Appendix A). The trend in decreasing the number of marbles suggested a possible anxiolytic effect. Consequently, we decided to investigate this effect of NIA in a more specific test for anxiety.

Mice were recorded for 10 min during an EPM section. The video analysis revealed that NIA-treated mice spent more time in the open arm and increased the number of transitions (5 ± 2.6 vs. 8 ± 2.7) (Figure 10A,B).

## 4. Discussion

In contemporary industrial society, the escalating demand for psychological and pharmacological interventions to address stress-related issues is a pressing concern. This phenomenon transcends age, gender, and species boundaries, encompassing a wide range of individuals and even the well-being of pets. Current drug therapies often exhibit limited efficacy and carry substantial risks, including the development of tolerance and addiction.

One promising strategy for identifying novel treatments involves the clinical evaluation of drugs approved for other conditions. Historically, drug repurposing has yielded significant successes, as exemplified by GLP1 drugs initially indicated for diabetes but subsequently approved for obesity treatment. A comprehensive literature review has revealed distinctive anti-stress properties of NIA. If further research confirms these findings, the potential for repurposing NIA as an anti-stress agent warrants exploration.

In this study, we investigated the effects of oral NIA administration on the spontaneous behavior of adult male mice. Our results showed mild anxiolytic effects without any evident side effects. Since 1971, only 23 articles on NIA have been published and indexed as articles on PubMed. Many of these articles are related to the sedative effects and the recent reassessment of the possible use of this drug in autism. In some of these studies, the possibility emerged that NIA could be used as an anxiolytic [53,56]. However, there are currently no preclinical behavioral data to support this hypothesis.

Our work partially addresses this gap and advances the hypothesis that this compound may combine other fundamental characteristics for the management of stress and anxiety in both humans and animals.

In a previous study, we used video tracking analysis and multivariate statistics to demonstrate that extracts from the Tilia Tomentosa bud extracts can reduce anxiety in mice when added to drinking water [57]. Building on this approach, we conducted a new study to test the effects of NIA on 17 different behavioral variables in mice after 14 days of oral administration. These data, obtained from two behavioral tests (MT and EPM), describe the motor activity, apathy, curiosity, and stress responses of each individual mouse. The NIA treatment was effective without affecting water consumption or weight gain. The stability study confirmed the linearity of NIA response in the range of concentrations covering the NIA mother solution and NIA solutions administered to mice. The NIA solutions administered as drinking water to mice remained stable at room temperature for up to 4 days; the mother solution used for their preparation was stable during 4-month storage at 2–8 °C.

The stress test performed by exposing NIA to extreme pH under 60 °C heating for 5 h evidenced no impurity formation; specifically, the formation of nicotinic acid, a possible NIA-related substance, under both acidic and alkaline pH was ruled out. The obtained results evidenced the stability of the molecule, even at relatively harsh conditions, and validated the chosen administration procedure.

PCA analysis of these variables identified two distinct groups based on pharmacological treatment. NIA treatment did not affect motor performance, including speed, acceleration, and distance traveled. However, mice showed a tendency towards sedation based on the number and duration of animal pauses, but this did not affect their exploration and curiosity. Our data agree with the previous literature focused on human clinical studies, which have shown the efficacy of NIA in children with sleep disorders. In a double-blind study, ten parameters (such as sleep latency, nighttime sleep motor activity, total sleep time, etc.) were evaluated, and mothers’ interviews were collected. The results provided strong evidence that NIA improved all ten parameters considered. In a group of 25 people with autistic disorder and associated behavioral alterations, an open-label study was conducted to evaluate the efficacy of NIA. In this study [53], the administration of a daily dose of NIA (1 mg/kg) for 60 days was followed by a positive effect in 52% of patients, particularly in hyperkinesia, unstable attention, resistance to change and frustration, mild anxiety signs, hetero-aggressiveness, and sleep disorders. NIA was more effective in subjects with mild or moderate mental retardation, and no adverse effects were observed. The literature underscores that NIA treatment could be short for most adolescents, and no adverse effects were observed [52,56]. It is worth noting that although NIA has been used in Europe, it does not have FDA approval for pediatric insomnia in the United States [58]. Interestingly, NIA treatment also led to a reduction in perceived stress and anxiety, as demonstrated by lower levels of risk-taking behavior (time in open arms) and fewer buried marbles. These effects align with previous studies on different histaminergic antagonists [59,60,61]. Indeed, the two receptor antagonists, chlorpheniramine and ranitidine, displayed anxiolytic effects in the EPM [39,62]. Additionally, H1 receptor knockout mice exhibited a prolonged transfer latency in EPM trials, indicating anxiolytic-like effects. Yanai et al. also showed that the absence of H1 receptors reduced aggressive-like behavior [63]. Currently, hydroxyzine is the only FDA-approved antihistamine to treat anxiety. Its efficacy, which in some studies is comparable to that of benzodiazepines, suggests that H1 inhibition may be an interesting strategy [64]. However, it is important to consider the side effects of antihistamines, which can be debilitating in the elderly and people with cognitive disorders. Based on the pharmacodynamic characteristics of NIA, we cannot rule out the possibility that its anxiolytic effect is dependent on its affinity for α1 receptors. Indeed, it is known that Prazosin and ACH-000029, two α1 ligands, reduce anxiety-like behaviors [65,66,67].

In addition to its use in sleep and mood disorders, NIA has also been studied as a treatment for Autism Spectrum Disorders [27,68] and pediatric insomnia associated with anxiety or mood disorders, psychosis, aggression, or anticipatory anxiety associated with medical procedures [58]. In the field of autism, several mouse models have been tested using behavioral test batteries, including the MT and the EPM [69]. Our findings show that NIA significantly elevated the frequency of freezing episodes during an MT trial and augmented adaptive stress tolerance in an EPM paradigm. Notably, NIA, at the administered dose, and despite the inherent limitations of self-administration by mice, did not appear to substantially alter motor performance or exploratory behavior. This characteristic could be leveraged to modulate the stress response in scenarios where excessive sedation induced by other treatments might compromise therapeutic adherence [70,71,72,73] or cognitive performance. Moreover, the potential to mitigate stress while preserving the physical capacity for interaction could be exploited in veterinary medicine for animals exhibiting high stress levels [73,74,75]. While further investigation is required to fully explore these possibilities, the findings suggest a promising approach that could complement existing therapeutic strategies. To facilitate future synergistic applications, a comprehensive characterization of NIA’s mechanism of action will be essential.

## 5. Conclusions

Our study revealed anxiolytic effects in mice treated daily with NIA. The fact that NIA does not appear to alter curiosity under light sedation while maintaining the anxiolytic effect suggests a potential for repurposing this old drug. However, further studies are needed to understand the dose–response relationship, species susceptibility, and mechanism of action of this potentially stress-correcting treatment in animals. A primary limitation of this study is the imprecise daily NIA dosage due to water administration. To address this, we will explore alternative experimental designs, such as gavage, and compare our findings to a benzodiazepine. The spontaneous regimen was chosen to minimize animal stress, as indicated by the selection of behavioral tests. The selection of a spontaneous regimen was motivated by the imperative to minimize stress on the animals, as also demonstrated by the choice of the Marble Test. The reduction in the number of animals used, coupled with the employment of advanced statistical techniques such as multivariate analysis, is a further optimization of experimental procedures in line with the 3R principles.

## Figures and Tables

**Figure 1 biomedicines-12-02087-f001:**
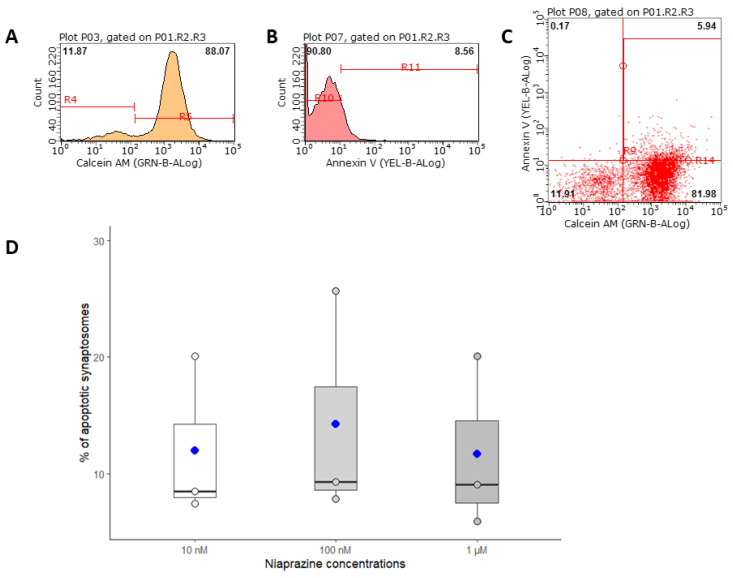
Cytofluorimetric analysis of cortical synaptosomes exposed to increasing concentrations of NIA for 1 h. Representative histograms of labeled synaptosomes with Calcein AM (**A**) and synaptosomes labeled with Annexin V (**B**). Representative double labeling of synaptosomes with Calcein-AM/Annexin-V PE and the percentage of apoptotic particles after 1 µM NIA (upper right corner) (**C**). Box plot representation of the mean of three experiments (n = 3 independent mice) performed in triplicate. Data are expressed as mean ± SEM (**D**). Analysis of variance was performed by ANOVA followed by Tukey’s multiple comparison test.

**Figure 2 biomedicines-12-02087-f002:**
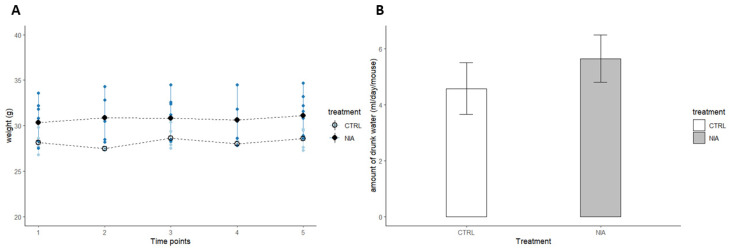
Body weight (**A**) and the amount of water consumed (**B**) were monitored during the treatment period. Data are expressed as mean ± SEM. Analysis of variance was performed by ANOVA followed by Tukey’s multiple comparison test. No significant difference was registered when comparing the body weight of control (CTRL) with that of treated (NIA) mice. Treated mice did not drink significantly more than control mice.

**Figure 3 biomedicines-12-02087-f003:**
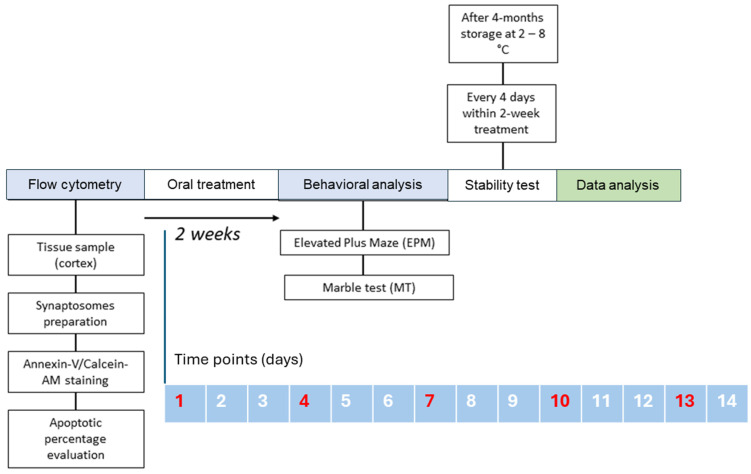
Experimental design of Niaprazine study based on ARRIVE 2.0 guidelines.

**Figure 4 biomedicines-12-02087-f004:**
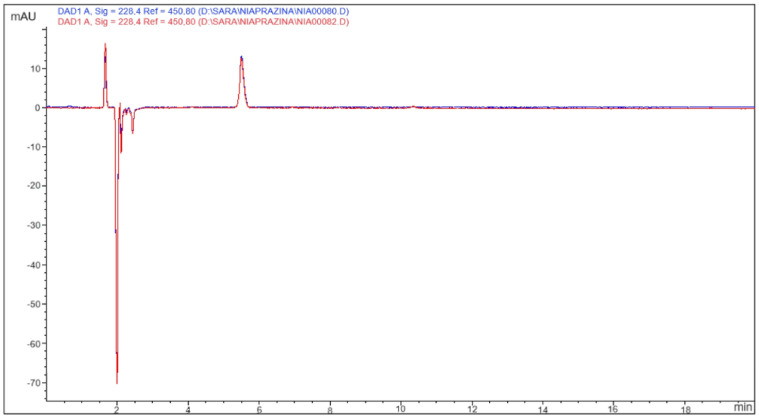
Representative chromatograms of NIA aqueous solutions after 4-day storage at room temperature.

**Figure 5 biomedicines-12-02087-f005:**
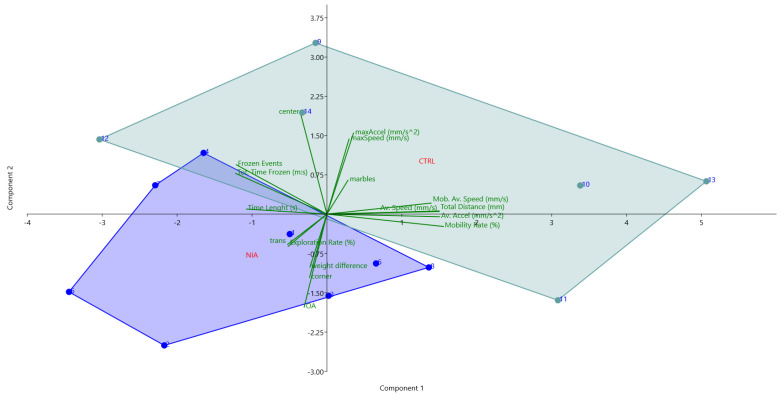
Bidimensional representation of Principal Component Analysis (PCA) shows the relationships among variables and two observed groups (the disposition of animals in the environment). Green arrows point in the direction of each variable. The control group was represented in light blue (n = 6) and the treated group in green (n = 8). Light blue dots represent each control mouse, while dark blue dots represent treated mice. Each group is described by a polygon.

**Figure 6 biomedicines-12-02087-f006:**
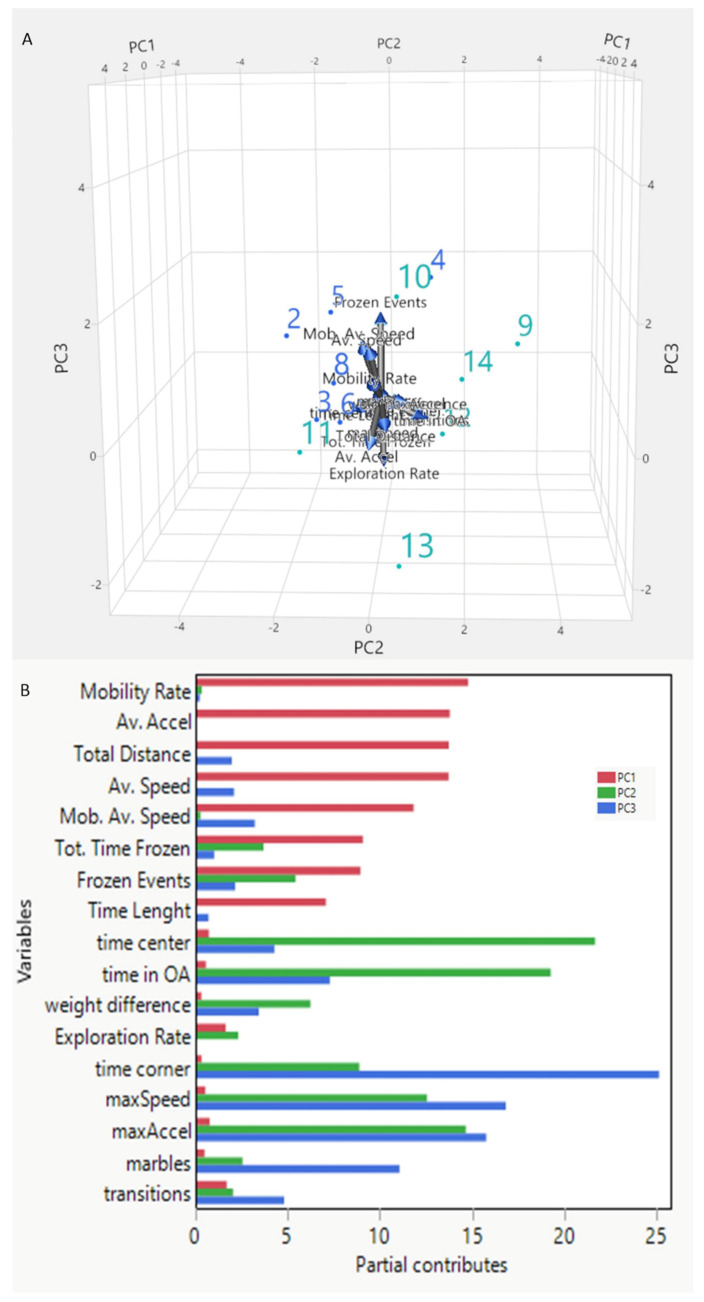
PCA analysis was performed using locomotor and exploratory parameters obtained from MT and EPM. Tridimensional representation shows the disposition of animals in the environment (**A**). Arrows point in the direction of each variable. The control group is represented in green and the treated group in light blue. (**B**) The contribution and strength of the variables to the principal components.

**Figure 7 biomedicines-12-02087-f007:**
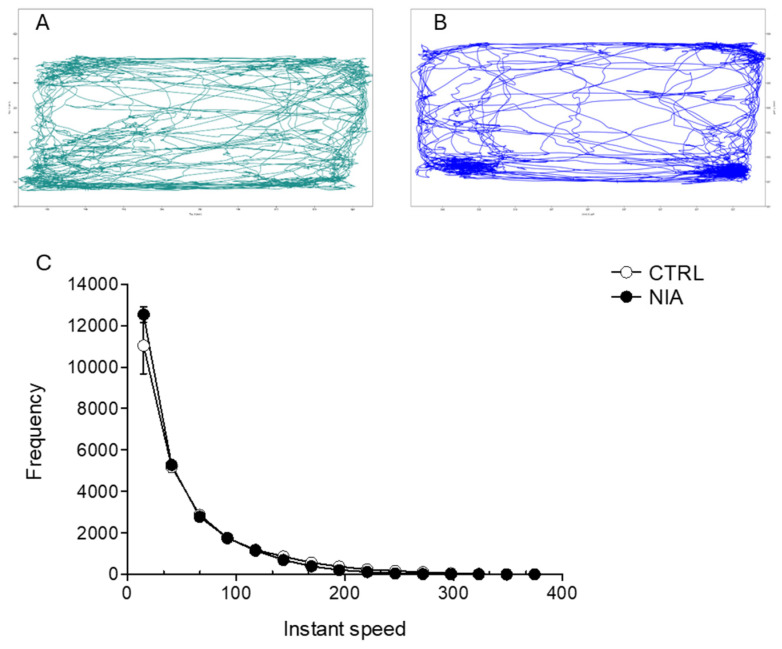
Representative trajectories for control and NIA-treated mice in the MT arena are displayed in (**A**,**B**), respectively. (**C**) Frequency of instantaneous speed. Analysis of variance was performed by ANOVA followed by Tukey’s multiple comparison test (n = 14).

**Figure 8 biomedicines-12-02087-f008:**
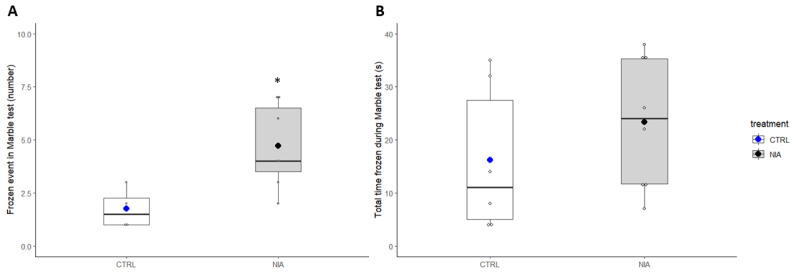
Immobility parameters during the Marble Test. Panel (**A**) shows the Frozen events (less than 3 mm movement in 5 s) in the control (white) and in the treated (grey) groups. Panel (**B**) illustrates the total time frozen in the control (white) and in the treated (grey) groups. Blue and black dots indicate the mean of control and treated groups, respectively; error bars indicate the standard error of the mean (SEM). The NIA group statistically increased the number of frozen events compared to the control group (**A**), *t*-test, * *p* = 0.021; there was no significant difference in total time frozen between the two groups.

**Figure 9 biomedicines-12-02087-f009:**
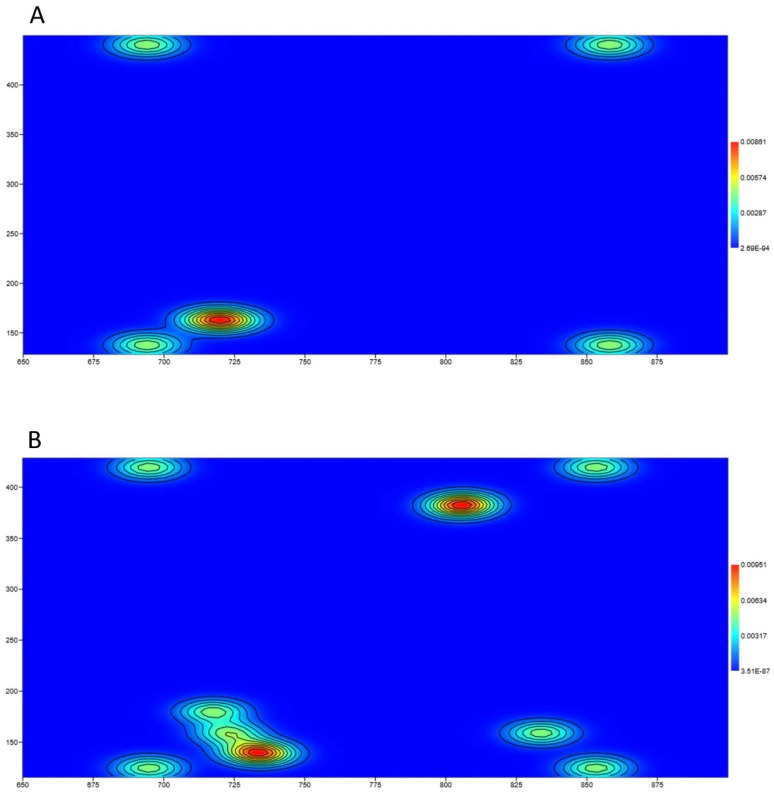
Representative heatmap in Marble Test. Each image shows preferred spots of frozen events in control (**A**) and NIA-treated (**B**).

**Figure 10 biomedicines-12-02087-f010:**
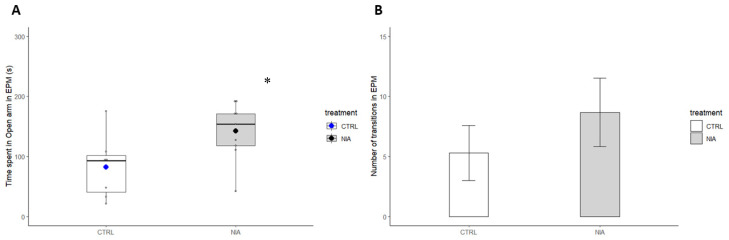
The time spent in the open arm was significantly higher in the NIA group compared to the control during EPM, represented by box plots (**A**). The number of transitions between the open and closed areas of the maze does not show statistical difference, represented by a bar chart (**B**). The white box and bar represent the control group (n = 6); the grey box and bar are used for the NIA group (n = 8). Data are expressed as mean ± SEM. * *p* = 0.025 vs. control, two-tailed Student’s *t*-test.

**Table 1 biomedicines-12-02087-t001:** Eigenvalue and percentage of variance explained by each variable (%).

Principal Component	Eigenvalue	Variance (%)
1	6.3883	37.578
2	2.65679	15.628
3	2.03895	11.994

## Data Availability

The original contributions presented in the study are included in the article; further inquiries can be directed to the corresponding authors.

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
