# Peer review of "Unveiling Niaprazine’s Potential: Behavioral Insights into a Re-Emerging Anxiolytic Agent"

_biomedicines, 2024, doi:10.3390/biomedicines12092087_

Round 1
Reviewer 1 Report
Comments and Suggestions for Authors
The article "Effect of Niaprazine chronic oral treatment on spontaneous behaviour of male mice: a preclinical study for anxiolytic evaluation", explores the potential therapeutic impact of Niaprazine (NIA) treatment as an anti-anxiety and anti-stress agent in a murine population. The in vivo treatment consisted of administration of NIA in the dinking water for 14 days. The effects of NIA on viability and apoptosis were assessed ex vivo in synaptosomes from murine brain cortex as an estimate of synaptic toxicity. Behavioural tests (plus maze test and maAnd rble test) were performed to estimate changes in levels of toxicity. The chemical stability of NIA was analysed.
There are inconsistencies in the experimental design
1) The modifications to the original method cited (citations 22 and 23) made to the isolation of synaptosomes are not indicated. Given that in these experiments the eventual toxicity of AIN will be determined, it would have been desirable to describe in detail how synaptic terminals were obtained, as a model for studying toxicity in terms of impact on both viability and apoptosis.
2) It is not understood why this approach was not also performed on synaptosomes from mice treated with NIA in drinking water, as they would have allowed to understand the impact of the compound in the conditions in which they were delivered and would have allowed to verify or refute the preliminary results obtained in (1) and to estimate the real impact of the doses of NIA delivered tin drinking water.
3) The experimental groups and the conditions of the treatments and controls are not all described in Materials and Methods, but some of them are described in Results. This leads to confusion and lack of clarity in the description of the experimental model.
4) In general, the "n" of the different experiments is either not indicated or is indicated with little clarity and precision.
5) It is striking that NIA is dissolved in miliQ water, given the differences in osmotic and salt content compared to the drinking water normally provided to mice.
6) The dose of NIA received via water intake is not precisely determined, including an additional variability factor, in the experimental design.
In terms of the results, significant differences were only observed between mice treated and not treated with NIA in frozen events (Immobility parameters, Fig. 9) and in the time spent in open arms in the EPM test (Figure 12), although the asterisk that could identify this, is not present.
The authors discuss the anxiolytic importance of NIA, based on these results, and considering that a 14-day treatment is sufficient to be considered "chronic". The authors contrast these results with others in the literature. We understand that they are not sufficient to determine the level of understanding of the impact that the authors attribute to NIA, which dangerously invades the realm of speculation.
Author Response
Reviewer 1
- The modifications to the original method cited (citations 22 and 23) made to the isolation of synaptosomes are not indicated. Given that in these experiments the eventual toxicity of AIN will be determined, it would have been desirable to describe in detail how synaptic terminals were obtained, as a model for studying toxicity in terms of impact on both viability and apoptosis.
Thank you for your comment. We agree and we have substituted the citation 23 with a more specific reference: Gylys KH, Fein JA, Cole GM. Quantitative characterization of crude synaptosomal fraction (P-2) components by flow cytometry. J Neurosci Res. 2000 Jul 15;61(2):186-92. doi: 10.1002/1097-4547(20000715)61:2<186::AID-JNR9>3.0.CO;2-X. PMID: 10878591.
Synaptosomes are well-established preparations for flow cytometry (see Gylys's group research). However, some limitations are well documented, and we have taken into consideration all the improvements suggested by Hobson BD, Sims PA in their article "Critical Analysis of Particle Detection Artifacts in Synaptosome Flow Cytometry" published in eNeuro in 2019. The preparation of crude synaptosomes is well-established, and therefore, we won't detail the methods to limit self-plagiarism.
- It is not understood why this approach was not also performed on synaptosomes from mice treated with NIA in drinking water, as they would have allowed to understand the impact of the compound in the conditions in which they were delivered and would have allowed to verify or refute the preliminary results obtained in (1) and to estimate the real impact of the doses of NIA delivered tin drinking water.
Following European legislation, particularly Italian law, we are required to minimize the suffering of animals used in research. To comply with this, we have conducted initial in vitro tests to avoid using toxic doses. The lack of preclinical data in the literature on NIA determined this choice. Throughout the treatment, we closely monitored the animals to assess their well-being. We believe that ex vivo experiments on treated animals can proceed, as long as there are no apparent toxic effects at the end of the treatment. Furthermore, in line with the guidelines of the animal ethics committee, we have conducted a washout to minimize the number of animals sacrificed.
- The experimental groups and the conditions of the treatments and controls are not all described in Materials and Methods, but some of them are described in Results. This leads to confusion and lack of clarity in the description of the experimental model.
We apologize for the lack of clarity. We have modified the methods accordingly.
- In general, the "n" of the different experiments is either not indicated or is indicated with little clarity and precision.
We apologize for the lack of clarity. We have modified the methods accordingly.
- It is striking that NIA is dissolved in miliQ water, given the differences in osmotic and salt content compared to the drinking water normally provided to mice.
We thank the Reviewer for underlining the mistake in line 181: actually, the solutions administered to the mice were prepared using drinking water. We apologize for this mistake, which has been amended in the revised version of the manuscript. As for the HPLC analyses, the standard solutions were prepared using milliQ water, which was also employed for the preparation of the mobile phase. However, no differences were observed in the HPLC chromatograms of standard solutions and analytes, and no additional peaks were present, so the use of drinking water in the analytes and milliQ water in the standard solutions has no impact on the outcome of the chromatographic analysis.
- The dose of NIA received via water intake is not precisely determined, including an additional variability factor, in the experimental design.
Certainly, we agree with the reviewer's assessment. This is indeed one of the major limitations of this paper. However, we can provide precise explanations. Our study aims to outline a low-stress treatment of NIA in mice. At the outset of this study, we were not aware of the additional properties of NIA. Furthermore, the existing literature did not mention stress-related effects for NIA. Gavage is the next step to assess the actual effectiveness of NIA as an anxiolytic, comparing it with a benzodiazepine or an antidepressant. We specify this limitation in the text (last sentence).
In terms of the results, significant differences were only observed between mice treated and not treated with NIA in frozen events (Immobility parameters, Fig. 9) and in the time spent in open arms in the EPM test (Figure 12), although the asterisk that could identify this, is not present.
We apologize for the lack of clarity. We have modified the figure accordingly.
The authors discuss the anxiolytic importance of NIA, based on these results, and considering that a 14-day treatment is sufficient to be considered "chronic". The authors contrast these results with others in the literature. We understand that they are not sufficient to determine the level of understanding of the impact that the authors attribute to NIA, which dangerously invades the realm of speculation.
Thank you for your comment. We steer clear of speculation, and our discussion aligns with this approach. However, we need to summarize our findings. The final sentence is quite clear: "However, further studies are needed to understand the dose-response relationship, species susceptibility, and mechanism of action of this potentially stress-correcting treatment in animals." Lastly, the debate regarding chronic, subchronic, or "not acute" is still ongoing. A 14-day treatment is not considered acute; therefore, we can use the term subchronic. However, some experts may disagree and argue that after 5 days, it should be considered chronic.
Reviewer 2 Report
Comments and Suggestions for Authors
Main error:
1 The animal groups in this manuscript should be divided into: blank control group, anxiety model group, anxiety model group+positive drug group (diazepam suspension group or other), anxiety model group+NIA (low, medium, and high dose groups). Thus, NIA's anti anxiety evaluation can be conducted.
2 The order of the charts and tables in this manuscript is chaotic, making it difficult for readers to understand the content of the manuscript.
Other errors:
1 What is the time unit of Fig3A?
2 I can't see the Fig7A image clearly.
3 The position of Fig12A * is incorrect, n=?
4 Fig11 should be deleted and explained in the discussion.
5 The title of the manuscript does not match the content.
6 The order in which the methods are presented is inconsistent with the order in which the results are presented.
7 What group of results are Fig2A, B, and C?
Comments on the Quality of English LanguageMinor editing of English language required.
Author Response
1 The animal groups in this manuscript should be divided into: blank control group, anxiety model group, anxiety model group+positive drug group (diazepam suspension group or other), anxiety model group+NIA (low, medium, and high dose groups). Thus, NIA's anti anxiety evaluation can be conducted.
Thank you for your comment. We appreciate the reviewer's suggestion for an experimental design for an anxiolytic. However, our work is primarily focused on the lack of preclinical information about NIA. As mentioned in the text, we initially evaluated NIA's antianxiety effects using the Marble test before proceeding to one of the most crucial anxiety tests, the EPM. In the initial stages of the study, we used the Marble test to assess the sedative, explorative, and motor effects of NIA on mice. Our objective was to determine the extent to which the sedative effects affected the mice's curiosity. Based on these results, we plan to move forward with a different approach, such as gavage, to precisely assess the dose-response curve alongside a standard drug.A sentence has been added to provide a clear explanation of this limitation.
2 The order of the charts and tables in this manuscript is chaotic, making it difficult for readers to understand the content of the manuscript.
According to the reviewer, we have modified some figures.
Other errors:
1 What is the time unit of Fig3A?
We apologize for the lack of clarity. We have modified the figure accordingly.
2 I can't see the Fig7A image clearly.
We have increased the size of the figure.
3 The position of Fig12A * is incorrect, n=?
We apologize for the lack of clarity. We have modified the figure accordingly.
4 Fig11 should be deleted and explained in the discussion.
We agree.
5 The title of the manuscript does not match the content.
We agree and we have modified the title of the manuscript
6 The order in which the methods are presented is inconsistent with the order in which the results are presented.
We apologize for the lack of clarity. We have modified the methods accordingly.
7 What group of results are Fig2A, B, and C?
We apologize for the lack of clarity. We have modified the legend accordingly.
Reviewer 3 Report
Comments and Suggestions for Authors
(1) Title: Generic compounds should be written in lower case (unless the journal's rules are different), therefore "..of niaprazine..."
(2) Line 45/46: "...does not recognise..." Consider using another term.
(3) Line 46: "induces" instead "induced"
(4) Figure 2D: in my opinion, the legend is not necessary (also, the order is different from the one in the figure). This applies also to other figures.
(5) Figure 3A. Please identify the time points. Are NIA-treated mice really not heavier if you analyse with a two-factor ANOVA? It is difficult to see/recognize what the small dots mean. I guess the individual values. Maybe it's better to have the both groups not on the same x-axis level but slightly shifted.
(6) Is figure 4 necessary?
(7) Results: I am not sure about the journal's politics but I would prefer to see the statistical values of the different analyses.
(8) I don't understand how Figure 10 helps. It's not necessary in my opinion.
(9) Line 309: Where is the statistics showing the NIA decreased the number of marbles. I cannot see this in the figure.
(10) In the discussion, I missed the discussion of the own data? What do the data really show? There is the statement "The NIA treatment was effective without ..." but what does this exactly means. Which effects do you believe to observe?
(11) Please consider limitations of our study.
Author Response
- Title: Generic compounds should be written in lower case (unless the journal's rules are different), therefore "..of niaprazine..."
Thank you for your comment. We have modified the title accordingly.
- Line 45/46: "...does not recognise..." Consider using another term.
Thank you for the suggestion. We have modified this sentence.
- Line 46: "induces" instead "induced"
We apologize for the error, we have corrected it.
- Figure 2D: in my opinion, the legend is not necessary (also, the order is different from the one in the figure). This applies also to other figures.
Thank you for your comment. We have removed the legend from figure 2D.
- Figure 3A. Please identify the time points. Are NIA-treated mice really not heavier if you analyse with a two-factor ANOVA? It is difficult to see/recognize what the small dots mean. I guess the individual values. Maybe it's better to have the both groups not on the same x-axis level but slightly shifted.
We apologize for the lack of clarity. We have modified the figure accordingly. The statistical analysis does not reveal statistical differences.
- Is figure 4 necessary?
We agree and we have eliminated this figure
- Results: I am not sure about the journal's politics but I would prefer to see the statistical values of the different analyses.
Absolutely, we agree.
- I don't understand how Figure 10 helps. It's not necessary in my opinion.
We can agree with the reviewer; however, it is important to evaluate the position of these events in line with an increase in immobility. As shown in the figure, mice are more static after NIA and the "safe zone" has increased, but it always remains in the periphery. Accordingly, we added a sentence to explain this concept.
- Line 309: Where is the statistics showing the NIA decreased the number of marbles. I cannot see this in the figure.
We agree that this is only a trend and the text must be corrected.
- In the discussion, I missed the discussion of the own data? What do the data really show? There is the statement "The NIA treatment was effective without ..." but what does this exactly means. Which effects do you believe to observe?
Thank you for this comment. We have modified the discussion.
- Please consider limitations of our study.
We have already considered the limitations but probably not in the correct way. Therefore, we have modified the text.
Round 2
Reviewer 1 Report
Comments and Suggestions for Authors
Regarding the reordering of data that should be in materials and methods and are in results, nothing has changed in this 2nd version with respect to the original version. For example, the data that appear from line 179 to line 188 have not changed, which I insist do not correspond to results but to data that should appear in the experimental design described in materials and methods.
Regarding the experimental “n” . In this new version, the “n” was incorporated in the figure feet, but not in the description of materials and methods to show in detail the experimental desing. For example in lines 120 and 121, as in the first manuscript, it is still indicated that the total “n” is 14, without specifying how many mice followed the treatment with NIA and which were the number of mice used as controls of such treatment. This is the unique reference to “n” of used mice in Materials and Methods. Except for this part, none of the other items described in Materials and methods indicate the “n” of the mice used. The description of the flow cytometry experiments again does not precisely state the number of animals used. “To fulfil the 3Rs, a limited number of animals were sacrificed for the preparation of synaptosomes” (line 69)
Concerning the response of authors about milliQ wáter: The remark I made about the use of milliQ water clearly referred only to the supply of NIA dissolved in it, given the harm that would be caused by SUPPLYING MICE with NIA dissolved in milliQ water. This is what was written in the first version of the manuscript. Clearly, my remark never referred to the use of MilliQ water for HPLC analysis, to which the authors now refer. However, the statement from line 179 that the drinking water supplied to the mice was made with milliQ water remains unchanged: “Young male mice received a dose of 1mg/Kg/day of NIA in their drinking water. Solutions had a concentration of approximately 14.5 μM and were prepared fresh by diluting 1 mL of an 842 μM stock solution in 50 mL with mQ water; drinking solutions were replaced every 4 days to reduce stress caused to the animals and minimize inter-day dose variability.”
Regarding de chronicity or subchronicity of NIA treatment: We understand that, the supply of NIA in water for 14 days corresponds to a subacute type of treatment, neither chronic nor subchronic, as was signaled in this 2nd versión of the manuscript by autor without any bibliographic reference. The toxicological criteria associated with repeated drug treatments with varying frequency in rodents can be characterized as: a) chronic, at least a period of 6 to 12 months; b) subchronic treatments with a duration of approximately 10% of the estimated lifetime of the animal model, which for a mouse could be 73 days considering a maximum lifespan of 2 years (730 days); and c) subacute treatments with a duration between 14 and 30 days (1) Chhabra R.S., Huff J.E., Schwetz B.S. and Selkirk J. A overview of precronic and cronic toxicity/carcinogenicity experimental study designs and criteria used by the National Toxicology Program, United States, 1990; 2) Piccirillo V.J. Repeated-Dose Toxicity Studies. In: Product Safety Evaluation Handbook: Ed. Gad S.C.; Ed. Marcel Dekker, Inc., United States, 201-222, 1999; 3) Bucher J.P., Portier C.J., Goodman J.I., Faustman E.M. and Lucier G.W. Worshop overview: National Toxicology Program Studies: Principles of dose selection and applications to mechanism based risk assessment. Fundamental Applied Toxicolgy, 31, 1-8, 1996).
Concerning the response about the analyse of synaptosomes from mice under treatment with NIA in drinking wáter or their controls:
The regulations that respect animal welfare are deeply rooted and widely applied in the world, particularly in the world of science. In this context, we must carry out experimental designs that guarantee the good use of animal resources. Therefore, the autor’s answer as to why they did not use the brains of the mice that drank water with NIA to obtain synaptosomes and thus be able to establish a more adjusted “dose/effect” relationship, does not fit the question I asked. The argument put forward by the authors of not performing such experiments due to compliance with the Italian animal welfare law reducing the use of mice, does not apply, since the animals used in the treatment with NIA and their controls, were anyway and finally sacrificed. At that time, their brains could have been removed and synaptosomes obtained, without affecting the total number of mice used. This would have made it possible to obtain very valuable information on the real impact of NIA dose that actually acted on the nerve synapses of these mice and to correlate it with levels of stress and anxiety and also, the eventual structural or functional modulation of NIA on them. This approach would have made it possible to apply the law of the three “R's” on which animal welfare regulations are based (Replace, Reduce, Refine). Since this experiment was not carried out, in order to answer this question, more mice will eventually have to be used and sacrified, which goes against the “reduction” in the number of animals to be used, which is contemplated in all international animal welfare regulations.
The conclusions is very bold, considering the meager results obtained. The authors have not answered most of the observations or have done so partially and insufficiently.
The content of the present manuscript could be a valuable contribution if it were deeply reformulated, with care and dedication so that the preliminary results shown on the NIA compound could be of value in a future therapeutic context. At present, the work cannot be published if aspects of content and presentation of the methodology are not contemplated and the interpretation of the obtained results is modulated downward, framing them in a more realistic context.
Author Response
Regarding the reordering of data that should be in materials and methods and are in results, nothing has changed in this 2nd version with respect to the original version. For example, the data that appear from line 179 to line 188 have not changed, which I insist do not correspond to results but to data that should appear in the experimental design described in materials and methods.
Dear reviewer,
I apologize for the mistake I made. My coauthors sent me the correct text for water dilution, but I mistakenly made changes to it. It was my fault. I disagree with your assessment that the original text was not changed. We modified the structure by moving text and figures according to three expert evaluations.
Regarding the experimental “n” . In this new version, the “n” was incorporated in the figure feet, but not in the description of materials and methods to show in detail the experimental desing. For example in lines 120 and 121, as in the first manuscript, it is still indicated that the total “n” is 14, without specifying how many mice followed the treatment with NIA and which were the number of mice used as controls of such treatment. This is the unique reference to “n” of used mice in Materials and Methods. Except for this part, none of the other items described in Materials and methods indicate the “n” of the mice used. The description of the flow cytometry experiments again does not precisely state the number of animals used. “To fulfil the 3Rs, a limited number of animals were sacrificed for the preparation of synaptosomes” (line 69)
Thank you again. We detailed the number of control (6) and treated (8) mice. The flow cytometry was performed in three independent mice and each condition (NIA concentration) was evaluated in triplicate. We incorporate this information in the material and methods.
Concerning the response of authors about milliQ wáter: The remark I made about the use of milliQ water clearly referred only to the supply of NIA dissolved in it, given the harm that would be caused by SUPPLYING MICE with NIA dissolved in milliQ water. This is what was written in the first version of the manuscript. Clearly, my remark never referred to the use of MilliQ water for HPLC analysis, to which the authors now refer. However, the statement from line 179 that the drinking water supplied to the mice was made with milliQ water remains unchanged: “Young male mice received a dose of 1mg/Kg/day of NIA in their drinking water. Solutions had a concentration of approximately 14.5 μM and were prepared fresh by diluting 1 mL of an 842 μM stock solution in 50 mL with mQ water; drinking solutions were replaced every 4 days to reduce stress caused to the animals and minimize inter-day dose variability.”
I apologize for the mistake and have modified the sentence according to my coworker's feedback.
Regarding de chronicity or subchronicity of NIA treatment: We understand that, the supply of NIA in water for 14 days corresponds to a subacute type of treatment, neither chronic nor subchronic, as was signaled in this 2nd versión of the manuscript by autor without any bibliographic reference. The toxicological criteria associated with repeated drug treatments with varying frequency in rodents can be characterized as: a) chronic, at least a period of 6 to 12 months; b) subchronic treatments with a duration of approximately 10% of the estimated lifetime of the animal model, which for a mouse could be 73 days considering a maximum lifespan of 2 years (730 days); and c) subacute treatments with a duration between 14 and 30 days (1) Chhabra R.S., Huff J.E., Schwetz B.S. and Selkirk J. A overview of precronic and cronic toxicity/carcinogenicity experimental study designs and criteria used by the National Toxicology Program, United States, 1990; 2) Piccirillo V.J. Repeated-Dose Toxicity Studies. In: Product Safety Evaluation Handbook: Ed. Gad S.C.; Ed. Marcel Dekker, Inc., United States, 201-222, 1999; 3) Bucher J.P., Portier C.J., Goodman J.I., Faustman E.M. and Lucier G.W. Worshop overview: National Toxicology Program Studies: Principles of dose selection and applications to mechanism based risk assessment. Fundamental Applied Toxicolgy, 31, 1-8, 1996).
Thank you for this message. I don’t agree with this classification when we don’t evaluate chronic toxicity, but the discussion could be long (see references below). Therefore, I eliminated the term chronic.
Poddar I, Callahan PM, Hernandez CM, Pillai A, Yang X, Bartlett MG, Terry AV Jr. Chronic oral treatment with risperidone impairs recognition memory and alters brain-derived neurotrophic factor and related signaling molecules in rats. Pharmacol Biochem Behav. 2020 Feb;189:172853. doi: 10.1016/j.pbb.2020.172853. Epub 2020 Jan 13. PMID: 31945381.
Caldarone BJ, Karthigeyan K, Harrist A, Hunsberger JG, Wittmack E, King SL, Jatlow P, Picciotto MR. Sex differences in response to oral amitriptyline in three animal models of depression in C57BL/6J mice. Psychopharmacology (Berl). 2003 Oct;170(1):94-101. doi: 10.1007/s00213-003-1518-7. Epub 2003 Jul 15. PMID: 12879206.
Sun B, Sterling CR, Tank AW. Chronic nicotine treatment leads to sustained stimulation of tyrosine hydroxylase gene transcription rate in rat adrenal medulla. J Pharmacol Exp Ther. 2003 Feb;304(2):575-88. doi: 10.1124/jpet.102.043596. PMID: 12538809.
Powell GL, Leyrer-Jackson JM, Goenaga J, Namba MD, Piña J, Spencer S, Stankeviciute N, Schwartz D, Allen NP, Del Franco AP, McClure EA, Olive MF, Gipson CD. Chronic treatment with N-acetylcysteine decreases extinction responding and reduces cue-induced nicotine-seeking. Physiol Rep. 2019 Jan;7(1):e13958. doi: 10.14814/phy2.13958. PMID: 30632301; PMCID: PMC6328917.
Café-Mendes CC, Garay-Malpartida HM, Malta MB, de Sá Lima L, Scavone C, Ferreira ZS, Markus RP, Marcourakis T. Chronic nicotine treatment decreases LPS signaling through NF-κB and TLR-4 modulation in the hippocampus. Neurosci Lett. 2017 Jan 1;636:218-224. doi: 10.1016/j.neulet.2016.10.056. Epub 2016 Oct 29. PMID: 27984197.
Galpern WR, Lumpkin M, Greenblatt DJ, Shader RI, Miller LG. Chronic benzodiazepine administration. VII. Behavioral tolerance and withdrawal and receptor alterations associated with clonazepam administration. Psychopharmacology (Berl). 1991;104(2):225-30. doi: 10.1007/BF02244183. PMID: 1652144.
Concerning the response about the analyse of synaptosomes from mice under treatment with NIA in drinking wáter or their controls:
The regulations that respect animal welfare are deeply rooted and widely applied in the world, particularly in the world of science. In this context, we must carry out experimental designs that guarantee the good use of animal resources. Therefore, the autor’s answer as to why they did not use the brains of the mice that drank water with NIA to obtain synaptosomes and thus be able to establish a more adjusted “dose/effect” relationship, does not fit the question I asked. The argument put forward by the authors of not performing such experiments due to compliance with the Italian animal welfare law reducing the use of mice, does not apply, since the animals used in the treatment with NIA and their controls, were anyway and finally sacrificed. At that time, their brains could have been removed and synaptosomes obtained, without affecting the total number of mice used. This would have made it possible to obtain very valuable information on the real impact of NIA dose that actually acted on the nerve synapses of these mice and to correlate it with levels of stress and anxiety and also, the eventual structural or functional modulation of NIA on them. This approach would have made it possible to apply the law of the three “R's” on which animal welfare regulations are based (Replace, Reduce, Refine). Since this experiment was not carried out, in order to answer this question, more mice will eventually have to be used and sacrified, which goes against the “reduction” in the number of animals to be used, which is contemplated in all international animal welfare regulations.
"I don’t agree with the reviewer. Firstly, the reviewer wrote this surprising sentence: "Animals used in the treatment with NIA and their controls were anyway and finally sacrificed." Why? Probably, the reviewer thinks …finally died. Otherwise, the reviewer doesn’t know the Italian legislation. The Italian “Ministero della Salute”, according to European legislation, encouraged the relocation of animals, avoiding euthanasia without scientific purposes (sacrifice). Moreover, washout programs are commonly used both for internal control and for limiting the number of animals.
But the reviewer raised a new important scientific point that can justify the sacrifice: "Is it possible to obtain valuable information on the real impact of NIA dose that actually acted on the nerve synapses of these mice and to correlate it with levels of stress and anxiety and also, the eventual structural or functional modulation of NIA on them." This is true! But the first (R1) question was why to perform cytofluorimetric analysis before and not also after the treatment? I responded that we performed some experiments to exclude synaptic toxicity at high concentrations of brain NIA. This was a precautionary assessment given the paucity of preclinical data in the literature.
Why don't you evaluate toxicity after the treatment? NIA is an already approved drug also used in children and animals that did not show evident alterations (eyes, hair, weight, faeces, urine).
Now the reviewer has asked us whether we can correlate ex vivo experiments with behavioral analysis. Considering the actual results, as mentioned in the conclusions, we will evaluate the comparative effects of NIA with a Benzodiazepine to evaluate the functional modulation of stress and anxiety.
"The conclusions is very bold, considering the meager results obtained. The authors have not answered most of the observations or have done so partially and insufficiently.
I appreciated the comment and I agree with the reviewer that the mistakes that I have made in the first answer is bothersome.
Conclusions are very bold!
We considered well-balanced the text but according to the referee we decided to change some words. Our study revealed anxiolytic effects in mice chronically orally treated with NIA. The fact that NIA does not appear to alter curiosity under light sedation while maintaining the anxiolytic effect suggests a potential for repurposing this old drug. However, further studies are needed to understand the dose-response relationship, species susceptibility, and mechanism of action of this potentially stress-correcting treatment in animals.
Reviewer 2 Report
Comments and Suggestions for Authors
Whether you need to prepare an anxiety model for MBT depends on your research purpose and design:
1. Evaluation of Medicinal or Interventional Effects: If you want to test the effects of certain drugs or therapeutic interventions on anxiety, you can perform the test on normal mice and observe whether these interventions alter the burying behavior.
2. Enhanced Anxiety Model: If you need to evaluate anxiety disorder models or want to test the effect of drugs under extreme anxiety conditions, you might need to create an anxiety model first through specific methods (such as stress exposure) before performing the MBT. This helps to simulate the anxiety conditions found in clinical environments and assess the effects of therapeutic approaches on severe anxiety states.
Comments on the Quality of English Language
Minor editing of English language required.
Author Response
Thank you once again for your comment. I completely agree with you, and our experimental design is compatible. I mentioned that anxiety was not the primary focus of our evaluation, and indeed, the Marble test is also a good way to evaluate compulsive-like behaviour. (Angoa-Pérez M, Kane MJ, Briggs DI, Francescutti DM, Kuhn DM. Marble burying and nestlet shredding as tests of repetitive, compulsive-like behaviors in mice. J Vis Exp. 2013 Dec 24;(82):50978. doi: 10.3791/50978. PMID: 24429507; PMCID: PMC4108161.) NIA has also been studied as a treatment for Autism Spectrum Disorders [6,46]and therefore we decided to evaluate in a MBT arena this drug. We have tested the animals in a MBT using also a computerized tracking system to evaluate motor activities. After the results combining tracking system and animal observation we decided to analyse more specific anxiety test. Subsequently, we now have a foundation to comparatively explore the effectiveness of NIA as an anxiolytic agent. One approach we plan to take is to elevate stress levels by implementing a restraint test before the MBT evaluation.